# Alcohol Use Disorder: Neurobiology and Therapeutics

**DOI:** 10.3390/biomedicines10051192

**Published:** 2022-05-21

**Authors:** Waisley Yang, Rohit Singla, Oshin Maheshwari, Christine J. Fontaine, Joana Gil-Mohapel

**Affiliations:** 1Island Medical Program, Faculty of Medicine, University of British Columbia, Victoria, BC V8P 5C2, Canada; waisleyyang@alumni.ubc.ca (W.Y.); rsingla@student.ubc.ca (R.S.); 2Psychiatry Residency Program, Faculty of Medicine, University of British Columbia, Victoria, BC V8W 3P5, Canada; oshin.maheshwari@alumni.ubc.ca; 3Division of Medical Sciences, University of Victoria, Victoria, BC V8W 2Y2, Canada; cjf275@gmail.com

**Keywords:** addiction, alcohol, alcohol use disorder, neurotransmitter, pharmacological therapy, non-pharmacological intervention

## Abstract

Alcohol use disorder (AUD) encompasses the dysregulation of multiple brain circuits involved in executive function leading to excessive consumption of alcohol, despite negative health and social consequences and feelings of withdrawal when access to alcohol is prevented. Ethanol exerts its toxicity through changes to multiple neurotransmitter systems, including serotonin, dopamine, gamma-aminobutyric acid, glutamate, acetylcholine, and opioid systems. These neurotransmitter imbalances result in dysregulation of brain circuits responsible for reward, motivation, decision making, affect, and the stress response. Despite serious health and psychosocial consequences, this disorder still remains one of the leading causes of death globally. Treatment options include both psychological and pharmacological interventions, which are aimed at reducing alcohol consumption and/or promoting abstinence while also addressing dysfunctional behaviours and impaired functioning. However, stigma and social barriers to accessing care continue to impact many individuals. AUD treatment should focus not only on restoring the physiological and neurological impairment directly caused by alcohol toxicity but also on addressing psychosocial factors associated with AUD that often prevent access to treatment. This review summarizes the impact of alcohol toxicity on brain neurocircuitry in the context of AUD and discusses pharmacological and non-pharmacological therapies currently available to treat this addiction disorder.

## 1. Introduction

Alcohol use disorder (AUD) can be defined as a chronically relapsing disorder characterized by the compulsion to ingest alcohol, the loss of control in limiting alcohol intake despite adverse health, social, and occupational consequences, and the emergence of a negative emotional state that can involve feelings of anxiety, irritability, and dysphoria when access to alcohol is prevented, reflecting a state of motivational withdrawal [1]. The 5th Edition of the Diagnostic and Statistical Manual of Mental Disorders (DSM-V), which was published in 2013, has integrated the previously used terms of alcohol abuse and alcohol dependence into a single condition referred to as alcohol use disorder (AUD). This is measured on a scale of severity ranging from mild to severe, depending on the number of diagnostic criteria met by the patient. There are many factors that influence a person’s susceptibility to alcohol addiction, including age at the onset of consumption, genetic predispositions including family history of AUD, as well as stress and other environmental and socioeconomic factors.

Additionally, chronic alcohol consumption has been shown to reduce total sleep time as well as quality [2].

AUD is a serious health condition, and alcohol in general is considered one of the leading preventable causes of death in the United States [3], where 14.4 million adults (ages 18+) and over 400,000 adolescents (ages 12–17) have experienced AUD [4]. Globally, the harmful use of alcohol causes approximately 5.9% of all deaths annually, and 5.1% of the global burden of disease is attributable to alcohol consumption [5].

Chronic exposure to alcohol has profound effects on multiple systems throughout the human body, including the cardiovascular, gastrointestinal, and nervous systems [6]. For the purposes of this review, effects outside of the nervous system are briefly described here. For example, heavy alcohol consumption significantly increases the risk of hypertension, atherosclerosis as well as all forms of stroke [7,8,9,10,11]. Furthermore, alcohol use leads to liver cirrhosis and a range of liver diseases, from liver fibrosis to alcoholic hepatitis [12,13]. Outside of the liver, chronic alcohol consumption can lead to other types of gastrointestinal diseases, including cancers [14,15] as well as acute and chronic pancreatitis [16,17]. Of note, AUD can also alter gut microbiota, which in turn can result in neuroinflammation [18,19].

While alcohol consumption can lead to serious psychosocial dysfunction as well as increased incidence of violence, intimate partner aggression, and suicide [20,21,22,23], prolonged alcohol use and alcohol addiction can also have long-term consequences on the brain and other body systems. Acute alcohol consumption leads to short-term alterations in neurological function primarily due to its actions on inhibitory neurotransmission. Whereas repeated consumption of alcohol over time leads to long-term changes in the functioning of several key neural circuits, causing a compulsion to consume this substance despite adverse consequences as well as the development of a negative emotional state when access to alcohol is restricted [1]. These alterations, among others, are characteristic of AUD and are also commonly associated with addiction to drugs other than alcohol.

Given this background and so as to effectively treat AUD, it is imperative that we understand the neurobiological mechanisms behind the development of addiction. In this review, we discuss the current literature on the neurobiology of AUD, with a focus on the biological changes that occur in the brain resulting in addiction. We also highlight the current non-pharmacological and pharmacological therapies available for the treatment of AUD and conclude by listing potential future directions in this rapidly evolving field of research.

## 2. Addiction and the Brain

Addiction is a dynamic dysregulation of the motivational circuits within the brain caused by exaggerated incentive salience and habit formation, deficits in reward function leading to increased stress, and compromised executive functioning. Addiction can be divided into three major stages involving specific neurocircuits within the brain: (i) basal ganglia-driven binge and intoxication, (ii) withdrawal and negative affect involving the amygdala, and finally, (iii) prefrontal cortex-driven preoccupation and anticipation [24] (Figure 1). Alterations to brain circuitry, including molecular and neurochemical changes that occur within these circuits, occur during the transition from controlled drug intake to addiction and influence vulnerability to relapse [1,25]. In addition, genetic modulation and epigenetic alterations within the brain circuits responsible for reward, such as the mesolimbic dopamine (DA)-ergic pathway, are also thought to influence the susceptibility to develop an addiction [26,27].

During the development of addiction, individuals move from impulsive to compulsive drug taking, which is accompanied by a shift from positive to negative reinforcement [28]. While impulsivity is the predisposition toward unplanned reactions to internal and external stimuli without regard for consequences to oneself or others [29], compulsivity is manifested by repetitive behaviours that are often excessive and inappropriate, conducted to reduce tension or anxiety from obsessive thoughts [30]. It has been well documented that alcohol withdrawal results in symptoms such as tremors, seizures, autonomic hyperactivity, vomiting, nausea, anxiety, and dysphoria, which contribute to the development of compulsivity, thus encouraging alcohol-seeking behaviours so as to reduce the malaise experienced by withdrawal. Of note, the processes of impulsivity and compulsivity are not mutually exclusive and can present in the same stage of addiction [15]. Activation of the stress response during acute drug intake, sensitization during repeated withdrawal, and persistence in protracted abstinence contribute to compulsive behaviours seen in addiction. Koob and Le Moal have proposed an allosteric model to explain the persistent changes in motivation that occur during addiction [25,31]. According to this model, addiction can be conceptualized as a cycle of increasing dysregulation of the brain reward and antireward systems, resulting in a negative emotional state that contributes to the compulsive use of drugs. Within the brain, the counter-adaptive processes that limit reward function are unable to return to the normal homeostatic range, leading to prolonged dysregulation affecting motivation and promoting drug-seeking behaviours in an individual.

Neuroimaging studies have frequently implicated the orbitofrontal cortex and anterior cingulate gyrus in the later stages of addiction, showing activation of these brain regions during intoxication, craving, and bingeing, and their inactivation during withdrawal [32]. As these regions are involved in higher-order functions such as modulation of salience value of reinforcers and control/inhibition of prepotent responses, alterations to the functioning of these regions are likely to increase susceptibility to developing an addiction.

### Stages of Addiction and Its Neuroanatomical Correlates

(i).Binge/Intoxication Stage—Basal Ganglia

The binge/intoxication stage of addiction results in the dysregulation of the brain circuits involved in the ability to mediate salience value, leading to the development of excessive drug-taking habits due to increases in DA neurotransmission during drug intake. Specifically, alcohol intoxication causes the release of DA and opioid peptides into the ventral striatum, an area of the brain implicated in reward valuation [33,34,35]. On the other hand, a fast and steep release of DA is required for the activation of low-affinity DA D1 receptors, which are necessary for the rewarding effects and triggering conditioned responses [36,37,38]. As such, drugs that cause addiction (such as alcohol) are capable of emulating the increase in DA that is triggered by a phasic DAergic firing, which matches the firing frequency associated with rewarding stimuli [39].

Protracted exposure to addictive drugs can trigger neuroadaptations in basal ganglia circuits, and such modifications are hypothesized to play a central role in the development of compulsive drug-seeking habits and vulnerability to relapse [40,41]. In addition, addictive drugs influence synaptic plasticity within the mesocorticolimbic DAergic system, as they specifically increase DA levels within the mesocorticolimbic circuitry [41]. Moreover, addiction also causes a glutamatergic imbalance within the corticostriatal pathways, further affecting reinforcement-seeking behaviors [42]. During this initial stage of addiction, opponent processes are also triggered, and these result in a decrease in reward function accompanied by increased brain stress. These processes appear to involve multiple neurotransmitter systems and their modulators, including serotonin (5-HT) [43], DA [44], various opioid peptides [33], acetylcholine (ACh) [45], gamma-aminobutyric acid (GABA) [46], and glutamate (Glu) [24,41].

(ii).Withdrawal/Negative Affect Stage—Amygdala

The withdrawal/negative affect stage is characterized by increases in stress and anxiety-like responses resulting from withdrawal from drugs and may involve emotional pain, malaise, dysphoria, and loss of motivation for natural rewards [24,47]. This stage is characterized by an elevation of the reward threshold during withdrawal, which appears to be highly correlated with an escalation in drug intake, as demonstrated by multiple animal studies [48,49]. Imaging studies have also shown a decrease in the ability of natural rewards to stimulate the reward circuit in the human brain, suggesting that in addiction, the perceived value of drug-related stimuli is enhanced at the expense of stimulation from natural sources of reward [38,50].

During this stage of the addiction process, significant “within-system” adaptations occur, which are aimed at neutralizing the effects of the drug and persist after cessation of drug use, leading to feelings associated with withdrawal. In the case of AUD, decreases in the activity of the DAergic and serotonergic neurotransmitter systems, which are seen in the mesolimbic circuit and the nucleus accumbens (NA) during withdrawal, suggesting that deficits in monoamine release may contribute to the development of negative affect and influence alcohol-seeking behaviours [51,52,53]. A major adaptation is a reduction in DAergic neurotransmission in the NA, which is also accompanied by a decrease in striatal DAergic responses [34,54,55,56,57]. Reduced DAergic function is hypothesized to decrease motivation for non-drug-related stimuli and lead to increased sensitivity to the drug in use [58]. Ventral tegmental area (VTA) GABA neurons have also been heavily implicated in alcohol reinforcement and reward, with studies suggesting that alcohol may increase DA activity via inhibition of GABA release onto DAergic neurons in the VTA [59,60]. Other alterations include changes to µ-opioid receptor responsivity, enhanced Glu *N*-methyl-D-aspartate (NMDA) receptor sensitivity in the NA, and increases in extracellular Glu levels in the NA [61,62,63].

As tolerance and withdrawal develop corticotropin-releasing factor (CRF)-, norepinephrine-, and dynorphin-mediated signalling are activated in the extended amygdala, contributing to the development of the negative emotional state experienced during this stage of the addiction process [28,64]. Indeed, prolonged alcohol exposure has been shown to produce persistent upregulation of CRF as well as CRF_1_ receptors in the brain, and this may contribute to withdrawal behaviours and influence withdrawal rates [65,66]. In agreement with this hypothesis, administration of a CRF antagonist has been shown to reduce alcohol self-administration in dependent animals, further implicating CRF dysregulation in the development of addiction-related behaviours [67,68]. In addition, the norepinephrine system within the extended amygdala has also been implicated in the development of addiction behaviours, as blockage of β-adrenergic receptors and administration of the α1 receptor antagonist prazosin have been shown to prevent alcohol withdrawal symptoms and suppress increased drinking associated with acute withdrawal, respectively [69,70]. Dynorphin/ĸ-opioid system dysregulation has also been associated with dependence and are thought to contribute to increased drinking associated with alcohol withdrawal [71,72,73]. Furthermore, alcohol withdrawal has also been shown to cause decreased levels of cyclic-AMP response element-binding protein (CREB) phosphorylation in the central and medial nuclei of the amygdala [74].

Neuropeptide Y (NPY), nociceptin, and endocannabinoids are endogenous anti-stress modulators that can buffer the brain against stress and influence vulnerability to developing addiction [28,75,76]. Of note, alcohol withdrawal has been shown to decrease the expression of NPY, a mechanism that may play an important role in alcohol-related withdrawal symptoms [77]. Importantly, given the ability of NPY to block the motivational aspects of dependence, NPY has emerged as a potential pharmacological target for the treatment of stress-related disorders and may also be useful in targeting the negative affect symptoms associated with alcohol abstinence and withdrawal [78,79,80]. On the other hand, nociceptin is a polypeptide related to dynorphin A that acts as the endogenous ligand for the nociceptin receptor (NOP). Nociceptin/orphanin treatment has been shown to decrease alcohol consumption, attenuate alcohol-seeking behaviour, reduce expression of somatic withdrawal signs, and reverse anxiety-like behaviours associated with chronic and acute alcohol intoxication [75,81]. As such, the NOP system is also a promising pharmacological target for alleviating alcohol withdrawal-related symptoms. Substantial evidence has implicated endogenous cannabinoids (eCBs) in mediating various forms of short- and long-term plasticity in brain regions associated with the etiology of addiction [82,83,84,85,86]. This plasticity is disrupted following exposure to drugs such as alcohol and may influence features associated with addiction, including increased behavioural and neurochemical responses to drug exposure, decreased extinction of memories related to drug use, enhanced cue-induced drug craving, and increased levels of stress responsivity, anxiety, and depression [76].

(iii).Preoccupation/Anticipation Stage—Prefrontal Cortex

The preoccupation/anticipation stage of addiction consists of the return to drug-seeking behaviours after a period of abstinence. During this stage, inappropriately heightened sensitivity to drug-related cues works in combination with low reward function and increased stress system activity, leading to pathological drug-seeking behaviours in individuals afflicted by addiction disorders. Moreover, dysregulation of key prefrontal cortex (PFC) areas responsible for executive control over motivation and goal-oriented behaviour influences the ability to inhibit prepotent responses to drug-related cues, further increasing vulnerability to addiction [87,88]. In support, human imaging studies have shown activation of PFC regions, including the dorsolateral PFC, anterior cingulate gyrus, and medial orbitofrontal cortex, in response to cues associated with addiction [89,90,91,92].

Dysregulation of multiple neurotransmitter systems has been linked to the diminished ability to control and inhibit drug-seeking behaviours. For example, the loss of Glu homeostasis is thought to impair prefrontal regulation of striatal circuitry, thereby impacting the control of drug-seeking behaviours [42]. Furthermore, an imbalance and dysfunction of DAergic circuits associated with emotional control and decision making has also been implicated in the loss of control leading to compulsive drug use, as revealed by neuroimaging studies [93]. Furthermore, recruitment of a subset of GABAergic and CRF-expressing neurons within the medial PFC during withdrawal and functional disconnection of the PFC from the central nucleus of the amygdala appears to be critical mechanisms in the loss of executive control in addiction [94]. Interestingly, grey matter volume deficits within specific regions of the frontal cortex have been shown to be predictive of the risk of relapse, potentially serving as a useful neuroanatomical marker of relapse risk and treatment outcomes in AUD [95].

In summary, addictive drugs act on multiple circuits within the brain, including those responsible for executive control, motivation, and reward, leading to a loss of inhibitory control, deficits in decision making, changes to reward and motivation, and increased activity of stress response systems. The symptoms of negative affect experienced during withdrawal encourage alcohol-seeking behaviours, and functional deficits of the brain render an individual more vulnerable to relapse even after abstinence from alcohol.

## 3. AUD and the Brain

### 3.1. Effect of Alcohol on Neurotransmitter Systems

Alcohol (ethanol) has a simple chemical structure that allows it to freely diffuse across the lipid bilayer of cell membranes. As such, alcohol molecules can directly interact with components of the cell membranes, such as receptors and transporters, as well as with several intracellular molecules and structures, thus impacting multiple cellular processes and functions. In particular, alcohol is able to alter synaptic function by impacting multiple neurotransmitter systems, including 5-HT, DA, GABA, Glu, and ACh (Figure 2). The following paragraphs briefly summarize some of the main effects of alcohol on these neurotransmitter systems. A more detailed overview of how alcohol impacts neurotransmission can be found elsewhere [96,97,98,99].

5-HT has been implicated in anxiety, depression, bipolar disorders, obsessive compulsive disorders, eating behaviour and obesity, and drug addiction [100,101]. Alcohol has been shown to potentiate the activity of 5-HT_3_ receptors in murine models using comparable alcohol concentrations to those seen in humans afflicted with AUD [102,103]. Furthermore, genetic variants linked to 5-HT_3_ receptor sensitivity have been shown to result in an enhanced DAergic reward pathway in humans [104]. In addition, alcohol dependence has been associated with changes in the transcription of the serotonin transporter (5-HTT), which is encoded by the *Slc6a4* gene and is responsible for controlling the pattern and magnitude of 5-HT activity [101,105]. Within the *Slc6a4* gene, a repeat element of variable length in the 5′ region and a single nucleotide polymorphism (SNP) in the 3′ untranslated region were shown to influence alcohol dependence and severity of drinking as well as response to 5-HT-targeted therapies in AUD patients, respectively [106,107]. Within this scenario, interactive effects of multiple sequence variations at different levels within a specific serotonergic pathway have been proposed to confer greater susceptibility to developing AUD when compared to single variations [108].

DA is known to play a central role in the development of drug addiction, with animal studies suggesting that alcohol administration causes enhanced DAergic neurotransmission within the VTA and a consequent increase in DA levels in the NA [109,110,111]. In AUD, reduced DA receptor sensitivity is thought to decrease motivation for endogenous effectors of the reward circuitry, leading to enhanced compensatory alcohol consumption [112]. Of note, various genetic mutations and polymorphisms that play a role in DAergic neurotransmission have been suggested to contribute to increased vulnerability to alcohol addiction, including the DA receptor D2 *Taq1A* polymorphism [113,114,115], the DA transporter gene *Slc6a3* polymorphism [113,116], and the missense mutation within the catechol-O-methyltransferase (*Comt*) gene [112,117,118,119]. However, further research is still required to completely elucidate the relationships among genetic factors, DAergic neurotransmission, and the development of AUD.

The endogenous opioid system has important implications for addiction, including modulation of DA release in the NA and of DAergic neurotransmission within the mesolimbic pathway [120]. Polymorphisms of the *Oprm1* gene, which encodes the µ-opioid receptor, have been studied in relation to alcohol addiction with mixed results [121,122,123,124,125,126]. Additionally, both the delta and kappa opioid receptors have also been implicated in alcohol addiction [127,128]. Indeed, single nucleotide polymorphisms of *Orpk1* and *Orpd1* genes may influence behavioural responses to naltrexone [127].

ACh is a neurotransmitter with a wide range of functions both within and outside the central nervous system. SNPs within the cholinergic receptor muscarinic-2 (*CHRM2*) gene have been associated with predisposition to alcohol and drug dependence and with the development of affective disorders, including major depressive disorder [129,130]. SNPs within the cholinergic receptor nicotinic alpha-5 subunit (*CHRNA5*) gene have also been associated with alcohol dependence [131].

The eCB system function is also affected by alcohol both acutely and chronically [132], and this system likely plays a complex role in addiction and withdrawal. Acutely, alcohol decreases levels of the eCBs Anandamide (AEA) and 2-arachidonoylglycerol (2-AG) in hippocampal, amygdala, PFC, and cerebellar tissue [133,134,135]. Long-term exposure to alcohol has been documented to reduce both the binding to and expression of the cannabinoid receptor type a (CB1) in the brain [136,137,138,139]. In some cases, these effects can be transient and are not evident after a period of abstinence from alcohol [136,137]. Further research is required in this area in order to better understand how the eCB system is affected by alcohol, as this system has the capacity to influence other neurotransmitter systems responsible for addiction in the brain.

GABA is the principal inhibitory neurotransmitter in the adult human central nervous system. Studies have shown that alcohol allosterically modulates GABA_A_ receptors, and this mechanism may contribute to tolerance, dependence, and withdrawal in AUD [140,141,142]. The sensitivity of GABA_A_ receptors to alcohol has been suggested to be regulated by phosphorylation of the gamma-2 subunit by protein kinase C (PKC) [143,144]. Disruption of PKCɛ, in particular, appears to disrupt voluntary drinking behaviour in mouse models [145,146]. Alcohol has been shown to enhance DAergic neuronal firing rate via decreased firing frequency of GABAergic units within the VTA and NA, thereby reinforcing the effects of alcohol within the pathways involved in reward [147]. In addition, other studies have shown that alcohol increases GABAergic neurotransmission in the cerebellum, hippocampus, and thalamus [148,149,150]. Furthermore, some studies have suggested a potential link between the presence of specific haplotypes within the *GABRA2* gene responsible for encoding the α2 subunit of the GABA receptor and susceptibility to developing AUD [151,152,153,154].

Glu is the major excitatory neurotransmitter in the human brain. Acute alcohol exposure generally inhibits Glu neurotransmission, whereas chronic exposure and acute withdrawal have the opposite effect [155]. Alcohol likely affects Glu neurotransmission by altering the function of both metabotropic (mGluRs) and ionotropic (iGluRs) Glu receptors. Upregulation of the metabotropic glutamate receptor 5 (mGluR5)-Homer2-phosphoinositide 3-kinase (PI3K) signalling pathway by binge drinking has been hypothesized to predispose toward a high binge-like alcohol-drinking phenotype [156]. In addition, abnormal hyperactivation of Ras-extracellular signal-regulated kinase (ERK) downstream of mGluR5 results in a hyper-glutamatergic state and has been thought to be a key factor in behaviours associated with addiction [157]. Alcohol drinking was also shown to attenuate the function of D_2_ DA autoreceptors and group II mGluRs within the posterior VTA [158]. On the other hand, alcohol has inhibitory effects on iGluRs, being capable of inhibiting NMDA receptors [159,160,161]. However, chronic alcohol exposure was shown to increase postsynaptic NMDA receptor function in the rat basolateral amygdala [162]. Of note, the relationship between both the NR2A and NR3A NMDA receptor subunits and susceptibility to addiction has also been investigated, with studies showing a role for these subunits in alcohol dependence and acute NMDA receptor sensitivity to alcohol [163,164]. Variations in the NMDA-dependent α-amino-3-hydroxy-5-methyl-4-isoxazolepropionic acid (AMPA) receptor trafficking cascade controlling Glu-related excitatory neurotransmission have also been associated with alcohol dependence [165]. Alcohol has also been shown to reduce NMDA receptor expression and function in the NA and to cause deficits in NMDA receptor-dependent long-term depression (LTD) in this brain region after protracted withdrawal [166]. In addition, chronic intermittent ethanol exposure (CIEE) was shown to affect kainate receptors and result in postsynaptic increases in Glu neurotransmission [167] while also increasing the amplitude and frequency of AMPA-receptor-mediated spontaneous excitatory postsynaptic currents in the rat basolateral amygdala [162]. Of note, microinjection of the AMPA-receptor antagonist 6,7-dinitroquinoxaline-2,3-dione (DNQX) was capable of attenuating withdrawal-related anxiety-like behaviours, suggesting that increased Glu function may contribute to anxiety during withdrawal from chronic alcohol exposure [162]. A more detailed review of the effects of alcohol on Glu reward circuitry can be found elsewhere [168].

Acute and chronic alcohol exposure has also been shown to affect synaptic plasticity, therefore influencing the efficacy of synaptic transmission at synapses. As explained above, alcohol can directly impact the major excitatory (i.e., glutamatergic) and inhibitory (i.e., GABAergic) neurotransmitter systems within the adult central nervous system, thus effectively contributing to changes in both long-term potentiation (LTP) and LTD, and influencing learning and memory processes [169,170]. Of note, pre-natal alcohol exposure has also been shown to have profound effects on hippocampal synaptic plasticity during development [171].

### 3.2. Effects of Alcohol on Other Synaptic Targets

Alcohol has been shown to interact both directly and indirectly with additional synaptic and intracellular signalling targets within the brain, and this topic has been reviewed elsewhere [172]. In this section, we will present a brief summary of the main effects of alcohol on some of the synaptic and molecular targets within the brain and how these can affect synaptic activity.

Small (SK) and large conductance (BK) Ca^2+^ and voltage-gated K^+^ channels have been implicated in alcohol tolerance and adaptive plasticity. Chronic alcohol exposure has been shown to reduce SK channel function in VTA DAergic and CA1 pyramidal neurons and disrupt the SK-channel-NMDA receptor feedback loop, contributing to alcohol-associated adaptive plasticity of glutamatergic synapses [173,174]. Chronic alcohol exposure also leads to enhanced intrinsic excitability and glutamatergic synaptic signalling in lateral orbitofrontal cortical neurons, a mechanism that may contribute to the impairment of behaviours associated with the orbitofrontal cortex in AUD [175], such as anxiety, impulsivity, and aggression [176]. Alcohol has also been shown to interact with BK channels; however, factors such as the level of the activating ligand (intracellular Ca^2+^), BK subunit composition, post-translational modifications, channel lipid microenvironment, and type of alcohol exposure determine whether or not potentiation or reduction in BK currents occur following alcohol exposure [177]. Alcohol has also been shown to activate G-protein-gated inwardly rectifying potassium (GIRK) channels [178], thereby regulating neuronal excitability and influencing the development of alcohol addiction [179].

In addition to influencing synaptic channels and receptors, there is some evidence that long-term exposure to alcohol may influence synapse structures. Binge alcohol exposure alters scaffolding proteins associated with excitatory synapses [180]. Notably, the morphology of synapses has been shown to be disrupted, and the sizes of dendritic spines are reduced by chronic alcohol exposure in utero, during adolescence, and adulthood in rodent models [181,182,183,184,185,186].

Noteworthy, chronic alcohol use has also been linked to changes in multiple intracellular signalling pathways that can affect synaptic function directly or indirectly. These include alterations in adenosine signalling [187,188], as well as changes in PKC and adenylate cyclase activity [189,190,191].

## 4. Factors That Predispose Patients to AUD

Multiple physiological, genetic, and environmental factors have been associated with an increased predisposition to developing an addiction. It is estimated that genetic factors may influence the susceptibility to develop AUD by 40–60% and impact aspects of this type of addiction such as the quantity of alcohol consumed, frequency of drinking, risk of toxicity, in addition to response to medications [192]. It is important to note that different social and environmental circumstances act on different genetic substrates, and multiple variations occurring within a given gene or set of genes can modulate the effect of single or multiple polymorphisms. In humans, variations within specific genes such as those encoding alcohol dehydrogenase (ADH) [193,194], aldehyde dehydrogenase (ALDH) [192,194,195], and corticotropin-releasing hormone receptor 1 (CRHR1) [196,197], have been identified that either grant protection or render an individual more vulnerable to addiction.

Importantly, various genetic factors can also influence the efficacy of several medications used to treat AUD. For example, the D4 dopamine receptor *Drd4-L* polymorphism has been shown to modulate the effect of naltrexone (see Section 5.2), whereas the presence of at least one Asp40 (G) allele in the µ-opiate receptor *Oprm1* gene has been associated with lower relapse rates and a slower return to heavy drinking following naltrexone treatment when compared to those homozygous for the Asn40 allele [198,199,200,201].

The age at which consumption begins plays a major factor in the development of addiction. Experimentation and addiction often start during adolescence, a period characterized by important developmental changes in the brain [202,203]. Drug exposure during adolescence has been associated with more chronic and intense use and a greater risk of developing a substance use disorder when compared to individuals who initiate use at an older age [203,204,205,206,207]. In addition, heavy alcohol use during adolescence has been associated with a range of neurocognitive and behavioural deficits, such as impairments in attention, memory, and visuospatial processing [208,209]. On the other hand, older adults have a higher risk of disability, morbidity, and mortality from many alcohol-related chronic illnesses [210,211], in addition to increased alcohol-drug interaction side effects, thereby influencing susceptibility to treatment [212,213].

Numerous environmental factors have been consistently associated with the propensity to use drugs and alcohol. Although a detailed review of environmental factors that can contribute to the development of addiction and AUD is outside of the scope of this review, it is worth listing here some of the more prominent factors, such as socioeconomic status and the availability of support systems, proximal factors such as parental drug and alcohol use habits, early life adversities, quality of parenting, parental mental health, influence by siblings and others, as well as distal factors such as availability of drugs, neighbourhood and school characteristics, advertising, and media influence [214,215]. AUD is also more prevalent among certain groups than others. The National Epidemiological Survey on Alcohol and Related Conditions III found that AUD was more prevalent among American men regardless of severity, among those who had been previously married or never married, and those with lower income, which again speaks to the fact that high socioeconomic status and the availability of a support system are powerful protective factors against addiction in general and AUD in particular. In addition, Indigenous respondents had higher rates of severe AUD when compared to Caucasian respondents [216], a fact known to be related to social determinants of health and the legacy of colonialism within the United States.

## 5. Treatment of AUD

There are several treatments currently available for AUD; however, access to treatment remains an issue. Although AUD is very common, with 5.9% of global deaths attributable to alcohol use [5], only approximately 22% of patients are receiving treatment for this damaging condition [5,217]. In the U.S., only 1 in 6 adults reported ever having their drinking behaviours assessed by a health care professional, and in 2015, only 8.3% of people received specialty treatment out of the 15.8 million adults reporting a need for treatment related to alcohol use [218,219]. Despite the high prevalence of AUD, the stigma associated with addiction and insufficient systematic screening in primary healthcare settings continue to pose barriers to seeking and receiving treatment [220]. In addition, access to treatment in the U.S. is limited to those with the ability to make time to seek care and to those who can financially afford it. Gender disparities also exist when it comes to AUD treatment, as women with AUD experience more barriers when seeking treatment and are less likely to access treatment when compared to men with AUD [221]. When AUD is left untreated, the resulting functional impairments have been associated with diminished opportunities, increased stressful life conditions, and increased risk for psychiatric disorders even after AUD remission, making it imperative to address both impaired functioning and alcohol consumption during AUD treatment [216,222].

Treatment for AUD includes both non-pharmacological interventions, such as motivational interviewing, cognitive behavioural therapy (CBT), group therapies, and support groups such as Alcoholic Anonymous (AA), as well as pharmacological approaches, including drugs targeting some of the neurotransmitter systems affected by alcohol (Figure 3).

In mild AUD, it is recommended to start one or more non-pharmacological approaches before embarking on pharmacological treatment, whereas a combination of non-pharmacological and pharmacological therapy is recommended in more severe cases.

### 5.1. Psychological and Non-Pharmacological Therapies for AUD

Non-pharmacological interventions for the treatment of AUD range from individual approaches to extensive in-patient residential treatment and from more traditional approaches such as counseling to the use of modern technology. The short-term goals of most psychological interventions include support for abstinence or reduction in substance use, with health care professionals promoting adherence and participation in treatment, as well as acting as a source of positive encouragement and reinforcement. Long-term goals include enduring abstinence, or consequence-free drinking of low amounts of alcohol, and supporting the patient in overcoming the mental health and social problems arising from AUD.

A brief intervention is a common initial step in the non-pharmacological treatment of AUD, involving collaborative empowerment and support (often following the format of motivational interviewing) provided by the physician or health care professional so as to encourage the patient to change their behaviours [223,224]. Motivational interviewing in particular includes providing feedback to the patient on risks undertaken, stressing that the agent of change is the patient themselves, providing options on how to change, and discussing and agreeing on goals while remaining empathetic through all interactions [224]. These types of brief interventions have been used to treat AUD for over 30 years and have demonstrated a positive effect on reducing immediate alcohol consumption when compared to more extensive counselling. However, achieving long-term optimal outcomes may be unrealistic if only a brief intervention is offered [223,224,225].

CBT is another form of structured one-on-one psychotherapy used to treat AUD, which focuses on increasing awareness of the interplay between cognition, emotions, and behaviour [226]. The goal of CBT is to correct the maladaptive thought processes learned over time in order to change subsequent emotions and behaviours. This can be administered in person with a trained therapist, via self-guided materials, online via Internet or smartphone applications, or in group sessions [224,226]. Multiple meta-analyses and review articles have found the efficacy of CBT in improving perspectives on alcohol and adherence to treatment [224,226,227]. Several group therapy options also exist for the treatment of AUD, including 12-step programs such as AA, a form of treatment centred around camaraderie and spirituality, where people are supported by peers and mentors facing similar challenges [228]. The efficacy of AA and similar twelve-step facilitation (TSF) programs has been examined in a Cochrane review, finding that AA/TSF may be superior to other treatments in increasing the percentage of abstinence days, particularly in the long term. On the other hand, AA/TSF probably performs as well as other psychological treatments with regards to AUD-related consequences, addiction severity, and reducing the intensity of alcohol consumption [228]. In addition, 4/5 of the reviewed economic studies found substantial cost-saving benefits in using AA/TSF [228]. Of note, clinically delivered TSF interventions designed to increase AA participation lead to increased rates of continuous abstinence, an effect largely achieved by fostering increased AA participation beyond the end of the TSF intervention. However, AA opposes any therapeutic approach that does not endorse abstinence as its end goal [228,229]. Nevertheless, offering non-abstinent treatment goals to patients demonstrates a willingness to work with patients rather than imposing a standard goal, thus increasing the likelihood that a patient remains involved in treatment, increasing their chances of recovery, and reducing AUD-related problems [230].

For more severe forms of AUD, in-patient residential treatment options are available. Residential treatment involves in-patient care at an alcohol-free residential facility with support staff and licensed counsellors, social workers, physicians, other allied health care professionals, and peers. Depending on the facility, treatment can incorporate a diverse set of therapy options, including individual and group sessions, social work and training, and access to medical, psychiatric, and psychological services. Treatment outcomes using this modality may vary depending on the level of external and internal control held by the patient [231]. When subsequently supplemented with AA upon discharge from residential care, residential treatment has been shown to improve abstinence rates [232].

Nutraceutical treatment of AUD is a promising method by which the toxic effects of alcohol on the body may be ameliorated by reducing oxidative stress in the body [233,234,235]. Indeed, compounds such as S-adenosylmethionine, which influences levels of reduced glutathione in the body, may protect against mortality in alcohol-induced liver cirrhosis [236,237,238]. Nicotinamide adenine dinucleotide phosphate (NADPH) oxidases may be key mediators of alcohol-induced damage. Therefore, nutritional treatments that influence NADPH function or the capacity to metabolize acetaldehyde (such as taurine and pantethine) may have protective effects against alcohol-induced damage [239,240,241]. In addition, omega 3 fatty acid supplementation has also been found to have protective effects against alcoholic liver disease and may also influence drinking behaviour [242,243,244].

Although outside the scope of the present review, it is worth noting that other non-pharmacological approaches that may have therapeutic value in AUD include repetitive transcranial magnetic stimulation, transcranial direct current stimulation, and deep brain stimulation. For a more in-depth discussion of these therapeutic interventions, please see [245,246,247,248].

### 5.2. Pharmacological Therapies for AUD

In addition to psychological therapies, there are many pharmaceutical options currently available for the treatment of AUD. In the U.S., there are several Food and Drug Administration (FDA)-approved drugs that can be used in AUD, including disulfiram, naltrexone, and acamprosate, in addition to other promising off-label pharmacotherapy candidates such as nalmefene, baclofen, and topiramate. Unfortunately, despite the high prevalence, mortality, and economic costs of AUD, these medications are currently under-prescribed, with one study showing that only 9% of individuals needing treatment receive a single prescription of any FDA-approved medication, likely due to factors such as lack of access, financial restraints, or insufficient health insurance coverage [249].

Disulfiram is an FDA-approved irreversible inhibitor of ALDH, leading to increased levels of acetaldehyde in the body and precipitation of the aversive disulfiram-ethanol reaction, which is characterized by tachycardia, nausea, flushing, vomiting, sweating, hypotension, and palpitations. The presumed effectiveness of disulfiram in treating AUD is based on the patient’s aversion to these effects rather than on its direct pharmacological action [250,251,252,253,254,255]. Despite being available for years, the efficacy of disulfiram is still debatable. One meta-analysis study found that disulfiram was more successful than control treatment in open-labelled randomized control trials (RCTs). In these studies, control groups received either placebo, acamprosate, naltrexone, or no disulfiram, and success was defined as one of the following: total abstinence, proportion of abstinent days to treatment days, mean days of alcohol use, likelihood of no relapse, time to first heavy drinking day, or three or more weeks of consecutive abstinence [256]. Nevertheless, supervised ingestion of the drug to ensure medical compliance was associated with significant positive outcomes, a finding not replicated with unsupervised treatment. The main adverse effects observed drug-drug interactions, as well as dermatological, hepatic, cardiac, psychiatric, and neurological symptoms and neuroimaging findings. Despite these adverse effects, disulfiram has an acceptable risk profile, being generally safe when used according to recommendations [257].

Naltrexone is a pharmacological compound that has been found to reduce craving and the reinforcing effects of alcohol. It is a non-selective opioid receptor antagonist that reduces opioidergic activity, thereby modulating the rewarding effects of alcohol [258]. Naltrexone is available in oral and long-acting injectable formulations that can last up to one month. Naltrexone has been found to reduce the risk of drinking relapse at approximately 3 months, decrease the number of heavy drinking days and risk of heavy drinking, decrease the total amount of alcohol consumed, and reduce the levels of gamma-glutamyltransferase, an enzyme positively correlated with liver damage [259,260]. Oral naltrexone intake is associated with a range of side effects, including abdominal pain, decreased appetite, nausea and vomiting, daytime sleepiness, drowsiness, fatigue, lethargy, insomnia, somnolence, and weakness. Of note, the effects of extended-release naltrexone are comparable with those of oral formulations [260]. Nevertheless, despite its moderate side effects, naltrexone remains an effective and safe pharmacological strategy for treating AUD [260].

Acamprosate is another FDA-approved drug used to treat AUD that modulates Glu neurotransmission. The majority of studies examining the efficacy of acamprosate in treating AUD support its use despite reporting small to moderate effect sizes. Acamprosate is successful in decreasing the risk of drinking relapse and in increasing the cumulative duration of abstinence when compared to placebo [259,261]. Diarrhea is the main reported side effect [261]. Despite its moderate effects, acamprosate appears to be a safe and effective treatment option to support continuous abstinence after detoxification [261].

Nalmefene is an antagonist of µ- and δ-opioid receptors and a partial agonist of κ-opioid receptors. Although this compound has now been approved for use in Europe, its use remains controversial [262,263,264,265,266], as this is associated with a plethora of adverse effects that include nausea, vomiting, dizziness, insomnia, headache, fatigue, and somnolence [267], and the magnitude of its efficacy is still under debate.

Baclofen is a GABA_B_ receptor agonist approved by the FDA and used to reduce spasticity associated with neurologic disorders such as multiple sclerosis. Baclofen has been shown to significantly increase the time to first drinking lapse and the percentage of patients abstinent at the end point of the study while also resulting in a trend toward an increase in the percentage of abstinence days when compared to placebo [268]. Of note, the efficacy of a low dose of baclofen was higher than that of a high dose of this drug, which was accompanied by a decrease in tolerability, although the occurrence of serious adverse events was rare [268]. In addition, it is worth noting that the effects of baclofen were more significant when the daily alcohol use at baseline was higher [268].

Gabapentin is an FDA-approved anticonvulsant that is commonly used in the treatment of epilepsy and neuropathic pain. It inhibits presynaptic voltage-gated sodium and calcium channels, thereby preventing the propagation of action potentials and the release of various neurotransmitters, including Glu. Gabapentin may have a role in treating the symptoms associated with mild alcohol withdrawal and should be considered for the treatment of alcohol dependence when there are barriers preventing the use of other medications [269]. The most common adverse side effects associated with gabapentin are dizziness, somnolence, peripheral edema, and gait disorder [270].

Topiramate is another FDA-approved drug used in the treatment of seizure disorder that is also effective in preventing migraines and facilitating weight loss (when used in combination with phentermine). Topiramate is thought to have multiple mechanisms of action, including blocking voltage-dependent sodium channels [271], increasing GABA levels through receptor interactions [272,273], antagonizing NMDA receptors [274,275,276], modulating L-type calcium channels [277], inhibiting carbonic anhydrase isoenzymes [273,278], and modulating the phosphorylation state of various membrane proteins [279,280,281]. Topiramate was shown to result in a greater number of abstinent days and lower binge drinking frequencies when compared to placebo treatment [280]. Topiramate seems to have a greater effect when compared to naltrexone and acamprosate, which are more commonly prescribed in AUD [280]. Further research is needed to clarify the context in which treatment with topiramate will be most beneficial.

Other off-label medications used in the treatment of AUD include but are not limited to ondansetron, varenicline, sodium oxybate, antidepressants, aripiprazole, quetiapine, arginine vasopressin type 1B receptor antagonist ABT-436, mifepristone, citicoline, carbamazepine, and valproate. An in-depth description of these medications is outside of the scope of the present review but has been reviewed elsewhere [282].

## 6. Conclusions and Future Directions

The etiology of AUD is complex and multifaceted, involving changes in neuronal and synaptic function, neurotransmitter systems, and ultimately brain circuitry. The neurobiological changes induced by alcohol exposure result in alterations of the pathways involved in motivation and reward, executive decision making, affect, and the stress response. Currently, FDA-approved medications for the treatment of AUD modulate several of the neurotransmitter systems impacted by alcohol during the binge/intoxication stage, with the goal of neutralizing or reversing its effects on these systems and associated neuronal circuits and ultimately blocking the motivation to seek alcohol. Novel pharmacotherapeutic development should focus on reversing the motivational and reward deficits seen during the withdrawal and preoccupation stages of addiction and may involve targeting the dysregulated stress circuits, including those of CRF, NPY, glucocorticoids, dynorphin/κ opioids, nociceptin, and endocannabinoids [283]. Combined pharmacological and non-pharmacological treatment is recommended where appropriate in more severe cases of AUD. Furthermore, since AUD impacts not only brain function and associated behaviours, but also the quality of life of AUD-afflicted individuals and their families, AUD treatment must address not only the neurological and physiological aspects of alcohol addiction but also the psychosocial factors that contribute to addiction, so as to provide holistic care to these individuals and ensure their success as they embark in treatment and rehabilitation.

## Figures and Tables

**Figure 1 biomedicines-10-01192-f001:**
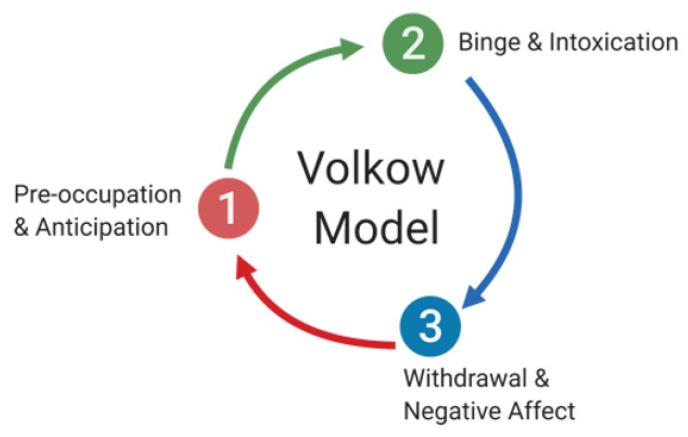
The cycle of addiction and its stages. Addiction can be divided into three major stages involving specific neurocircuits within the brain: (**1**) preoccupation and anticipation, which involve prefrontal cortex circuitry, (**2**) binge and intoxication, which is thought to primarily involve basal ganglia circuitry, and (**3**) withdrawal and negative affect, which are dependent on the amygdala.

**Figure 2 biomedicines-10-01192-f002:**
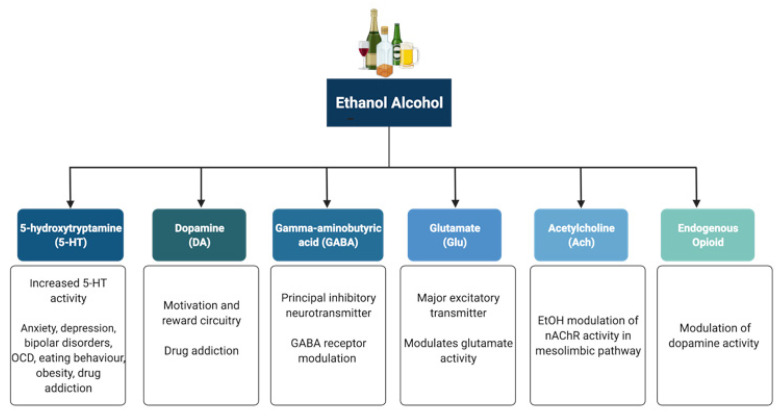
Neurotransmitter systems affected by alcohol (ethanol). Alcohol can interact with multiple neurotransmitter systems in the brain, including the serotonergic (5-HT), dopaminergic (DA), gamma-amynobutyric acid (GABA)-ergic, glutamatergic (Glu), Acetylcholinergic (ACh), and opioid systems, disrupting synaptic transmission and signalling and resulting in the dysregulation of neuronal networks that control reward, motivation, decision making, affect, and the stress response.

**Figure 3 biomedicines-10-01192-f003:**
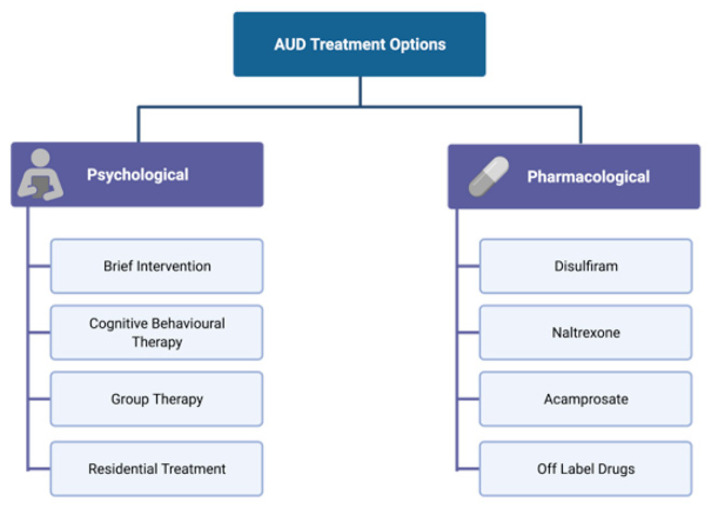
Psychological and pharmacological interventions for the treatment of AUD. AUD treatment includes both non-pharmacological (psychological) interventions that vary from individual approaches to extensive in-patient residential treatment, as well as several pharmacological approaches that target some of the neurotransmitter systems affected by alcohol.

## Data Availability

No new data were created or analyzed in this study. Data sharing is not applicable to this article.

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
