# Peer review of "Alcohol Use Disorder: Neurobiology and Therapeutics"

_biomedicines, 2022, doi:10.3390/biomedicines10051192_

Round 1

Reviewer 1 Report

Review of manuscript entilted: “Alcohol Use Disorder: Neurobiology and Therapeutics” authored by Waisley Yang, Rohit Singla, Oshin Maheshwari, Christine Fontaine, Joana Gil-Mohapel

First of all I want to thank you for opportunity to review this interesting manuscript.

Manuscript is well-written and easy to follow. Authors covered several aspects of alcohol dependence like its neurobiology and therapy. I think this review can be nice contribution in the field of dependence as a good source of basic knowledge.

Major concerns:

  • Regarding the opioid system in alcohol dependence, authors mentioned only the Oprm1 gene polymorphisms, while recent reports also show that Oprd1 and Oprk1 may also be crucial for this as reported here (doi: 10.1016/j.pbb.2012.08.019 and 10.1016/j.neuropharm.2017.03.016). Moreover in the second manuscript there is an interesting observation, which shows that naloxone (also the non-selective antagonist) efficiency relays on endogenous opioid system activity.

Minor concerns:

  • Line 8 on page 3 – opened bracket before citation
  • Line 23 on page 4 – not unified citation (Mitchell et al.)
  • Line 24 on page 4 – please order the citation numbers in ascending way
  • Line 25 on page 4 – numeration of subparagraphs is not proper
  • Line 41 page 8 – capital letter is missing (“of note”)
  • Line 19 page 9 – opened bracket before citation
  • Figures are a little bit blurry in my version of manuscript

Author Response

Reviewer #1

Manuscript is well-written and easy to follow. Authors covered several aspects of alcohol dependence like its neurobiology and therapy. I think this review can be nice contribution in the field of dependence as a good source of basic knowledge.

Major Concerns:

  1. Regarding the opioid system in alcohol dependence, authors mentioned only the Oprm1 gene polymorphisms, while recent reports also show that Oprd1 and Oprk1 may also be crucial for this as reported here (doi: 1016/j.pbb.2012.08.019 and 10.1016/j.neuropharm.2017.03.016). Moreover, in the second manuscript there is an interesting observation, which shows that naloxone (also the non-selective antagonist) efficiency relays on endogenous opioid system activity.

Response: We thank the Reviewer for pointing out this omission. We have now added a short blurb to the revised manuscript to address this point and have cited the references suggested by the reviewer. The new blurb reads as follows (see page 7, lines 35-37):

Additionally, both the delta & kappa opioid receptors have also been implicated in alcohol addiction [127-128]. Indeed, single nucleotide polymorphisms of Orpk1 and Orpd1 genes may influence behavioural responses to naltrexone [127]. 

Minor Concerns: 

  • Line 8 on page 3 – opened bracket before citation
  • Line 23 on page 4 – not unified citation (Mitchell et al.)
  • Line 24 on page 4 – please order the citation numbers in ascending way
  • Line 25 on page 4 – numeration of subparagraphs is not proper
  • Line 44 page 8 – capital letter is missing (“of note”)
  • Line 19 page 9 – opened bracket before citation
  • Figures are a little bit blurry in my version of manuscript

Response: We thank the Reviewer for their careful revision of our manuscript and for pointing out these minor errors. They have all been corrected in the revised version of our manuscript. In addition, we have supplied high-resolution versions of the figures to the editorial office to the editorial office.

Reviewer 2 Report

This review investigates alcohol use disorders and its implication with neurobiology. This kind of overview is useful for an high impact disease worldwide. The review is overall objective. Organization is slim for the subject. English could benefit from a review by a native speaker. Reading is set to an audience of specialists in neurology.

Before publication, some major issues have to be addressed:

1) Introduction in missing an overview on the effects of alcohol on other major organs mainly the heart and the liver which also have an effect on neurobiology. A figure is also in need to assemble visually the narrative of the review. 

2) Therapeutic considerations are missing nutraceuticals and their beneficial effects.

3) The conclusion is insufficient. The authors should propose a neurological rehabilitation program. The scope of the review must also be set to give new ideas for AUD.

Author Response

Reviewer #2

This review investigates alcohol use disorders and its implication with neurobiology. This kind of overview is useful for an high impact disease worldwide. The review is overall objective. Organization is slim for the subject. English could benefit from a review by a native speaker. Reading is set to an audience of specialists in neurology.

Response: We thank the Reviewer for his comments, which we have addressed below. We also would like to clarify that all Authors either speak English as their first language or are bilingual (please note that our group is located in British Columbia, Canada, where English is an official language).

Major Concerns: Before publication, some major issues have to be addressed:

1. Introduction is missing an overview on the effects of alcohol on other major organs mainly the heart and the liver which also have an effect on neurobiology. A figure is also in need to assemble visually the narrative of the review. 

Response: While the present review is focused specifically on the effects of alcohol on the brain, we have now added a paragraph to the Introduction summarizing some of the effects of alcohol consumption on other body systems. We have also cited in this paragraph a few detailed reviews on the effects of AUD on the body. This new paragraph reads as follows (see page 2 lines 6-15):

Chronic exposure to alcohol has profound effects on multiple systems throughout the human body including the cardiovascular, gastrointestinal and nervous systems [6]. For the purposes of this review, effects outside of the nervous system are briefly described here. For example, heavy alcohol consumption significantly increases the risk of hypertension, atherosclerosis as well as all forms of stroke [7-11]. Furthermore, alcohol use leads to liver cirrhosis and a range of liver diseases from liver fibrosis to alcoholic hepatitis [12-13]. Outside of the liver, chronic alcohol consumption can lead to other types of gastrointestinal diseases including cancers [14-15] as well as acute and chronic pancreatitis [16-17]. Of note, AUD can also alter gut microbiota, which in turn can result in neuroinflammation [18-19].

Following the Reviewer’s suggestion, we have now created a Graphical Abstract that summarizes the topics covered in our review article and have included this new Graphical Abstract/Figure in our revised manuscript.

2. Therapeutic considerations are missing nutraceuticals and their beneficial effects.

Response: We thank the reviewer for pointing out this omission. We have now added a paragraph summarizing some of the nutraceutical therapies that have been used to treat AUD to Section 5.1. The new paragraph reads as follows (see page 12, lines 31-40):

Nutraceutical treatment of AUD is a promising method by which the toxic effects of alcohol on the body may be ameliorated by reducing oxidative stress in the body [233-235]. Indeed, compounds such as S-adenosylmethionine, which influences levels of reduced glutathione in the body, may protect against mortality in alcohol-induced liver cirrhosis [236-238]. Nicotinamide adenine dinucleotide phosphate (NADPH) oxidases may be key mediators of alcohol-induced damage. Therefore, nutritional treatments that influence NADPH function or the capacity to metabolize acetaldehyde (such as taurine and pantethine) may have protective effects against alcohol-induced damage [239-241]. In addition, omega 3 fatty acid supplementation has also been found to have protective effects against alcoholic liver disease and may also influence drinking behaviour [242-244].

3. The conclusion is insufficient. The authors should propose a neurological rehabilitation program. The scope of the review must also be set to give new ideas for AUD.

Response: As discussed throughout the review, a single treatment (pharmacological or non-pharmacological) is, in most cases, short-sighted and likely insufficient. We have reworded the final paragraph of the Conclusion to highlight the fact that selecting treatment programs that not only influence the physiological effects of AUD but also impact the psychosocial aspects of addiction are likely to produce better and more long-lasting effects. The revised paragraph reads as follows (see page 14, lines 40-47):

Combined pharmacological and non-pharmacological treatment is recommended where appropriate in more severe cases of AUD. Furthermore, since AUD impacts not only brain function and associated behaviours, but also the quality of life of AUD-afflicted individuals and their families, AUD treatment must address not only the neurological and physiological aspects of alcohol addiction, but also the psychosocial factors that contribute to addiction, so as to provide wholistic care to these individuals and ensure their success as they embark in treatment and rehabilitation.

Round 2

Reviewer 2 Report

The authors have made the required adjustments to make the article fit for publication.